# Synthesis and Predicted Activity of Some 4-Amine and 4-(α-Aminoacid) Derivatives of N-Expanded-metronidazole Analogues

Justyna Żwawiak * and Lucjusz Zaprutko

Department of Organic Chemistry, Pharmaceutical Faculty, Poznan University of Medical Sciences, Grunwaldzka 6, 60-780 Poznan, Poland
* Correspondence: jzwawiak@ump.edu.pl; Tel.: +48-618-546-678

**Abstract:** The discovery of azomycin provided the major impulse for the systematic search for medicines showing activity against anaerobic protozoa. Nowadays, many other interesting applications have been found for nitroimidazoles as therapeutic agents. This research led to the acquisition of numerous new 4-amine-5-nitroimidazole derivatives, which have a structure analogous to metronidazole, characteristic of medicines most widely used in the treatment of anaerobic bacteria, protozoa and parasitic infections. The therapeutic activity of the described compounds is analyzed and confirmed with predictive methods.

**Keywords:** nitroimidazoles; N-alkyl-5-nitroimidazoles; nitro group ipso-substitution; aminonitroimidazoles; metronidazole analogues

## 1. Introduction

Many imidazole derivatives exhibit significant biological activity, which makes them widely used in many fields of pharmacology [1,2]. Moreover, the imidazole moiety is present in some natural biologically active compounds [3].

The nitro group is one of the most important groups that can appear on the imidazole ring. A lot of research concerns these compounds and their derivatives [4,5]. This is due to the activity of this class of compounds. Since the isolation of azomycin (2-nitroimidazole) [6], there has been a significant increase in interest in these substances. Many years of research have confirmed that most of the nitroimidazoles exhibit antiprotozoal and radiosensitizing activity [7,8]. The ability to radiosensitize is related to their one-electron reduction potential $E^{1/2}$, as well as to the oil/water partition coefficient. Suwiński et al. [9] studied the values of this coefficient for imidazole derivatives differing in the position of the nitro group and the presence of such substituents in the ring, i.e., –CH$_3$, –Cl, –Br, –OCH$_3$, –NH$_2$, or the second –NO$_2$ group. These authors proved that the lipophilicity of nitroimidazoles depends on the position of the nitro group in the molecule, and for similarly substituted compounds it decreases in the order of 5-nitro, 2-nitro, 4-nitro derivatives. They paid a lot of attention to the methyl substituent. Introducing it into the ring (at C-4 or C-5 position) increases lipophilicity, while introducing it into the N-1 atom causes an increase in the hydrophilicity of imidazoles. Moreover, the presence of the methyl group at the C-2 position has a significant influence on the ease of substitution reactions at the C-4 or C-5 positions of the imidazole ring [9].

5-Nitroimidazole derivatives are used primarily as antibacterial drugs. They are characterized by multidirectional action, which mainly includes absolutely anaerobic bacteria and protozoa [10–13].

The most commonly used drug in this group is metronidazole [14]. It is a substance with a wide range of applications, with limited toxicity at the same time [14]. Moreover, it is usually well tolerated by patients. Metronidazole has an antibacterial effect against many

clinically important microorganisms, including *Bacteroides*, *Clostridium* and *Fusobacterium*. In gynecology, it is known as an extremely effective agent in the treatment of bacterial vaginosis and trichomoniasis caused by the anaerobic protozoa *Trichomonas vaginalis*. The group of nitroimidazole derivatives also includes tinidazole [8]. Its spectrum of activity concerns bacteria of the genera *Gardnerella*, *Eubacterium*, *Campylobacter*, *Actinomyces*, *Spirochetes* and *Propionibacterium*. Other nitroimidazole derivatives are, among others, nimorazole used in acute, necrotic stomatitis and nonspecific vaginitis, and ornidazole, effective in the treatment of infections caused by bacteria from the *Trichomonas*, *Leishmania* and *Entamoeba* groups. It is also used to prevent perioperative infections [15]. As a member of the class of 2-nitroimidazoles, Misonidazole was previously considered in clinical trials as a radiosensitizer for the treatment of hypoxic tumors. Though tests revealed that it was a highly efficient radiosensitizer, the trials turned out to be unsuccessful due to the high neurotoxicity [16].

The aim of this work was to obtain a number of new 5-nitroimidazole derivatives with a potential biological effect. The expected activity may result from the structural analogy of the newly obtained compounds to substances mentioned above with known pharmacological activity. The compounds with a nitro group at position 5 are usually more active than the corresponding 4-nitro derivatives [1]. The introduction of an electron-accepting substituent at position 5 in the 4-nitroimidazole ring causes an increase in cytotoxic and radiosensitizing activity [17]. A number of azole derivatives with nitro and amino group in the same molecule are known to show antifungal, anti-HIV and antioxidant activity [1,18]. On the other hand, the length of the alkyl chain on the imidazole ring is relevant for antimicrobial activity. Khabnadideh et al. [19] showed that antibacterial activity of 1-alkylimidazoles against Gram-negative and Gram-positive bacteria increases as the number of carbons in the alkyl chain rises up to nine and then decreases.

## 2. Materials and Methods

2-Methyl-4(5)-nitroimidazole and 2-methyl-4,5-dinitroimidazole were obtained by the known procedures [20]. Commercially available chemicals were purchased from Aldrich (Springfield, MO, USA) and Chempur (Piekary Śląskie, Poland), and were used without further purification.

The melting points of the resulting series of compounds were measured in open capillaries with a Köfler apparatus and are reported without correction for the measurement conditions. Proton ($^1$H NMR) and carbon ($^{13}$C NMR) magnetic resonance spectra were recorded in DMSO-d$_6$. A Varian Gemini 300VT spectrometer (Varian, Palo Alto, Santa Clara County, CA, USA) with tetramethylsilane (TMS) as an internal standard was used to record the spectra. The chemical shift values ($\delta$) are expressed with an accuracy of $\pm 0.01$ ppm. Coupling constants are expressed in Hz. Mass spectrometry was recorded on an ADM 402 apparatus. Thin layer chromatography (TLC) was performed on aluminum plates covered with Kieselgel 60 DC-Alufol silica gel from MERCK (Darmstadt, Germany). This is a bottom-up technique using the development phase which was a mixture of $CH_2Cl_2$ and $CH_3OH$ in a volume ratio of 9:1. The spots of the synthesized products were visualized with UV light ($\lambda$ = 254 nm). The TLC plate shows an intensive yellow spot with an $R_f$ value higher than for the opposite isomer. Preparative column chromatography (CC) was performed by gravity using a 30 cm long and about 1.8 cm internal diameter chromatography column packed with 70–230 mesh silica gel from the MERCK company.

## 3. Results

Based on the earlier works [21–25], according to which a commercially available 2-methyl-4(5)-nitroimidazole can be nitrated to 2-methyl-4,5-dinitroimidazole (**1**) and a number of new aminonitroimidazole derivatives can be obtained, a series of compounds were synthesized, of which the nitro group at position 4 in substrate **1** was substituted with selected primary and secondary amines and, also via the amino group, derived from various α-amino acids. The general route of synthesis is shown in Scheme **1**.

**Scheme 1.** The general pathway of the described synthesis. Reagents and conditions: (**a**) EtOH, double excess of primary aliphatic /aromatic amine, $\Delta T$; (**b**) EtOH, double excess of secondary cyclic amine, $\Delta T$; (**c**) 75% EtOH, triple excess of α-aminoacid, $\Delta T$.

The first step was obtaining 2-methyl-4,5-dinitroimidazole. The basic substrate was 2-methylimidazole. This compound was nitrated according to the methods obtained from the literature, using a nitrating mixture consisting of concentrated nitric and sulfuric acids [20]. The reaction was carried out in two steps.

Then, the corresponding N-alkyl derivatives were formed during 60 min heating of 2-methyl-4,5-dinitroimidazole (**1**) and glycidyl isopropyl ether mixed in a molar ratio of 1:8, without the use of a solvent. After completion of the reaction, water was dropped to the mixture and allowed to stand for about 24 h. After this time, the crude product was precipitated, filtered off and purified by crystallization from ethanol: water mixture (4:6) with charcoal.

### 3.1. Reactions of N-Alkyl-2-methyl-4,5-dinitroimidazole (*2*) with Primary and Secondary Amines

Earlier works revealed that the substitution of the nitro group with amines differed in the 4 and 5 position of the 2-methyl-4,5-dinitroimidazole ring in an alcoholic solution [17]. It is worth noting that the substitution of the amines at the 4 position gives products a structure similar to metronidazole.

### 3.1.1. Reactions with Primary Amines

Firstly, the focus was on obtaining of new compounds with the use of primary amines, such as isobutylamine, aniline and p-bromoaniline. 3-Isopropoxy-1-(2-methyl-4,5-dinitroimidazol-1-yl)propan-2-ol (**2**) and the corresponding primary amine were mixed in a 1:2 molar ratio and dissolved in ethanol. Then, the solution was left to stand for 30 min at room temperature and extracted repeatedly with methylene chloride.

During the analysis of the crude obtained product, small amounts of the 4-nitroimidazole isomer were observed on the TLC plate, which, however, remained in the liquors during the crystallization process. Isomer 4-nitro can be characterized by lower value of $R_f$ in comparison with $R_f$ value of isomer 5-nitro. After the product obtained was isolated, it was crystallized from water to lead easily to the corresponding 4-amino-5-nitroimidazole derivative (**3**–**5**).

### 3.1.2. Reactions with Secondary Amines

In the next step of the research, an attempt was made to react the previously prepared 3-isopropoxy-1-(2-methyl-4,5-dinitroimidazol-1-yl)propan-2-ol (**2**) with secondary cyclic amines: morpholine, piperidine and N-methylpiperazine. The way of carrying out the reaction was analogous to the method described above: the starting materials were mixed in the ratio 1:2 and then dissolved in ethanol, mixed and left for 30 min at room temperature.

Obtained compounds **6**–**8** were not solidified and must be isolated by column chromatography using a mixture of methylene chloride and methanol (9:1). Small amounts of the 4-nitroimidazole isomer were isolated as impurities. Pure products **6**–**8** have been obtained as yellow solids.

### 3.2. Reactions of N-Alkyl-2-methyl-4,5-dinitroimidazole (**2**) with α-Aminoacids

In the next stage of the study, previously used compound **2** was reacted with α-amino acids. For this purpose, methionine and valine were used. In both cases, the substrates were combined in a 1:3 molar ratio (N-alkyl-2-methyl-4,5-dinitroimidazole: α-amino acid, respectively), then 75% ethanol was added and refluxed for 30 min.

Final products **9** and **10** were isolated by column chromatography on silica gel with a 9:1 mixture of methylene chloride and methanol. As impurities, small amounts of the unreacted substrate and 4-nitroimidazole isomer were present.

The identity and structure of the compounds obtained were confirmed by spectral analysis methods. The *m/z* values for the molecular ions of the substrate (**2**) and the products (**3**–**10**) are in agreement with the calculated molecular weights.

In the $^1$H NMR spectra of the derivatives obtained by the treatment of the primary aliphatic amine, isobutylamine, a singlet at 7.6 ppm was observed corresponding to one proton bound to the amino group. On the other hand, for compounds **4**, **5** containing an aromatic ring linked to an amino group, the corresponding signal appeared at about 8.5 ppm. This signal is used to distinguish two isomeric types of compounds, because in the $^1$H NMR spectrum of opposite isomer, the 4-nitro-5-amino-corresponding peak can be found at 8.7 ppm, therefore it is shifted about 0.2 ppm. This difference in the position of these signals can enable a simple distinction between the isomers obtained. The protons of the benzene ring in the system of aniline and its p-bromo derivative gave respective signals in the form of multiplets observed in the range of 6.7–7.3 ppm.

The peak located at about 5.3 ppm was assigned to the hydroxyl group. In the range of 3.2–3.9 ppm, a wide multiplet was visible indicating the presence of hydrogen atoms bounded to the alkyl chain substituted at the N-1 position of the imidazole ring. In a few cases (**3**, **6**, **9**, **10**), this range also included signals from some protons of amino or α-amino acid residues substituted in the C-4 position of the imidazole ring. The singlet at about 2.3 ppm corresponded to the methyl group bounded to the C-2 atom of the imidazole ring, while the peak at about 1.1 ppm corresponding to the six protons was assigned to two methyl groups derived from the isopropoxy moiety included in the N-1 alkyl chain.

In the $^{13}$C NMR spectra, signals from the carbon atoms of the imidazole ring were usually found at about 144 (C-4), 141 (C-2) and 128 (C-5) ppm. Three peaks appearing in the range 49–71 ppm were assigned to the carbon atoms of the aliphatic chain bounded directly to the N-1 atom of the imidazole ring. A signal at 50.7 ppm is assigned to a carbon atom connected with an oxygen atom from the isopropoxy part of the N-aliphatic chain. At about 22 ppm, a signal characteristic of the carbon atoms of the methyl groups of the isopropoxy moiety was visible, while the peak at about 14 ppm was connected with the C-2 methyl group. The remaining signals were from the carbon atoms of the corresponding amine or α-amino acid residues substituted at the C-4 position of the imidazole ring and their position corresponds to the expected shift values for this type of atom.

Detailed experimental data for all obtained products are described in Appendix A.

## 4. Discussion

*Calculation of the Level of Potential Biological Activities of the Obtained Compounds Using the PASS and AntiBac-Pred Methods*

In order to estimate the biological activity of chemical substances, we used, among others, the PASS (Prediction of Activity Spectra for Substance) program. This program is available on the website www.pharmaexpert.ru/PASSOnline/ (accessed on 13 November 2022). Based on the structure of the compound, it predicts with an appropriate probability the occurrence of a pharmacological effect, as well as the mechanism of action, mutagenicity, teratogenicity and embryotoxicity of this compound. The PASS program determines the probability of occurrence of a given activity (Pa) or its absence (Pi) for the analyzed chemical compound based on the probability of its presence in the set of active and inactive substances of the SAR (structure–activity relationship) set. The values of Pa and Pi are in the range 0–1 or 0–100%. If the Pa result is >70%, this compound is likely to exhibit this specific biological activity in vitro [26].

Additionally, we decided to predict the biological activity by the second in silico method for the obtained compounds. Using AntiBac-Pred web services of Way2Drug platform [27], activity against Gram-positive and Gram-negative bacteria was predicted for synthesized Metronidazole derivatives. This program is based on the data on antibacterial activity that are accessible in ChEMBL (a database of bioactive molecules with drug-like properties). It indicates the opportunity to classify chemical structures according to whether they inhibit or do not inhibit the growth of 353 distinct bacterial strains, including resistant and non-resistant strains [23]. It was revealed that the majority of these new products might be active against *Pseudomonas aeruginosa*-resistant strains, *Prevotella disiens* and *Bacteroides stercoris* strains with high probability value (>80). The best results were found for compounds with a secondary cyclic amines moiety in C-4 (**6–8**) with probability value >90. All the tested products are characterized by high antibacterial potential.

All nitroimidazole derivatives obtained in this study were analyzed by the PASS and AntiBac-Pred methods. Table 1 below shows the results of the predicted biological activities of the compounds obtained (**3–10**).

It was noticed that most of the products obtained here have a chance of showing the activity increasing the sensitivity to radiation. Moreover, the PASS program predicted antiprotozoal activity in all tested compounds (for reaction products with amines—activity of 60–70%, for reaction products with amino acids—50%). Moreover, the anti-alcohol activity that appeared with a probability close to 40% was determined for compound **3**. The substances containing in their structure fragments derived from α-amino acids (**9**, **10**) seem to be more promising as potential anti-alcohol drugs, for which the Pa values were about 50%. In the case of compounds **7–10**, the PASS program also predicts a cardioprotective effect at the level 65–70%, and compound **9** that may be effective in the treatment of myocardial ischemia also seems promising.

**Table 1.** The results of in silico analysis for products obtained **3–10**.

| No. | Pa | PASS Activity | Pa | AntiBac-Pred Activity |
|---|---|---|---|---|
| **3** | 0.752 | Radiosensitizing | 0.84 | Prevotella disiens |
| | 0.608 | Antiprotozoal | 0.84 | Bact. stercoris |
| | 0.520 | Ophtalmic drug | 0.76 | Clostridium ramosum |
| **4** | 0.673 | Radiosensitizing | 0.84 | Prevotella disiens |
| | 0.631 | Antiprotozoal | 0.83 | Bact. stercoris |
| | 0.529 | Skeletal muscle relaxant | 0.75 | Clostridium ramosum |
| **5** | 0.733 | Diazylglicerol lipase inhibitor | 0.78 | Bact. stercoris |
| | 0.658 | Radiosensitizing | 0.77 | Prevotella disiens |
| | 0.631 | Anti-infective | 0.68 | Clostridium ramosum |
| **6** | 0.702 | Radiosensitizing | 0.93 | Pseudomonas aeruginosa (resistant strains) |
| | 0.624 | Antiprotozoal | 0.91 | Prevotella disiens |
| | | | 0.82 | Bact. stercoris |
| **7** | 0.687 | Radiosensitizing | 0.92 | Pseudomonas aeruginosa (resistant strains) |
| | 0.633 | Cardioprotective | 0.86 | Prevotella disiens |
| | 0.617 | Antiprotozoal | 0.82 | Bact. stercoris |
| **8** | 0.686 | Radiosensitizing | 0.94 | Pseudomonas aeruginosa (resistant strains) |
| | 0.647 | Cardioprotective | 0.83 | Prevotella disiens |
| | 0.626 | Antiprotozoal | 0.80 | Bact. stercoris |
| **9** | 0.727 | Myocardial ischemia therapy | 0.78 | Bact. stercoris |
| | 0.718 | Radiosensitizing | 0.75 | Prevotella disiens |
| | 0.667 | Cardioprotective | 0.73 | Clostridium ramosum |
| **10** | 0.728 | Radiosensitizing | 0.80 | Bact. stercoris |
| | 0.675 | Cardioprotective | 0.80 | Prevotella disiens |
| | 0.565 | Antiprotozoal | 0.73 | Clostridium ramosum |

Pa—The probability of activity.

## 5. Conclusions

As shown by the results of our in silico biological studies, the products are part of a series of biologically active nitroimidazole derivatives, and they proved to be the important group of bioactive compounds. The described derivatives show especially strong antiprotozoal activity, so they can be considered as substances with potential pharmacological significance. Our results proved the important information on the effects of certain substituents and structural elements connected with the imidazole ring, and on the occurrence of certain activity or lack thereof, i.e., the presence of nitro groups and N-1 alkyl chains. Moreover, these studies can be helpful in the future planning of the syntheses of active drugs using nitroimidazole scaffold. Some modifications of the structure can influence the level of useful biological potencies.

**Author Contributions:** Conceptualization, L.Z. and J.Ż.; Methodology, J.Ż.; Investigation, J.Ż.; Writing—Original Draft Preparation, J.Ż.; Writing—Review and Editing, J.Ż. and L.Z.; Supervision, L.Z. All authors have read and agreed to the published version of the manuscript.

**Funding:** This research received no external funding.

**Data Availability Statement:** Not applicable.

**Conflicts of Interest:** The authors declare no conflict of interest.

## Appendix A. Experimental Data

*Appendix A.1. 3-Isopropoxy-1-(2-methyl-4,5-dinitroimidazol-1-yl)propan-2-ol (**2**)*

In a 100 mL round bottom flask, 0.42 g (2.5 mmol) of 2-methyl-4,5-dinitroimidazole (**1**) was mixed with 2.52 mL (20 mmol) of glycidyl isopropyl ether. It was heated in a water bath without the addition of solvent for 1 h. After the reaction mixture had cooled down, 40 mL of cold water was added and allowed to stand for 24 h. The separated dark orange material was filtered off and 0.54 g (77% yield) of crude product was obtained. Then, it was crystallized from water and ethanol mixture (6:4) with the charcoal. As a result, 0.41 g (57% yield) of a cream solid was obtained with mp = 104–106 °C, $R_f$ = 0.82 ($CH_2Cl_2$:$CH_3OH$ 9:1).

$^1$H NMR: δ 5.46 (d, *J* = 5.2 Hz, 1H, OH), 4.38–4.44 (m, 1H, N–$CH_2$), 4.13–4.21 (m, 1H, N–$CH_2$), 3.81–3.87 (m, 1H, CH–OH), 3.26–3.61 (m, 3H, O–$CH(CH_3)_2$ and $CH_2$–O), 2.51 (s, 3H, $CH_3$), 1.09 (s, 6H, $OCH(CH_3)_2$).

$^{13}$C NMR: δ 145.52 (C-4 Im), 139.38 (C-2 Im), 131.11 (C-5 Im), 71.39 (CH–OH), 69.30 (N–$CH_2$), 67.93 ($CH_2$–O), 46.69 ($OCH(CH_3)_2$), 21.89 ($OCH(CH_3)_2$), 13.63 ($CH_3$).

MS *m/z* (%): 288.3 M$^+$ (18.9).

HRMS (ES): calcd. for $C_{10}H_{16}N_4O_6$: 288.25740, found: 288.25772.

IR: 3270 (ν O–H, br); 2890 (ν C–H, m); 1524, 1326 (ν C–$NO_2$, m); 1485 (∂ C–H, m); 1385, 1365 (ν >$C(CH_3)_2$, m); 1170 (ν C–O–C, m); 870 (ν C–H, m).

*Appendix A.2. General Methods for the Reactions with Primary and Secondary Amines*

In a 100 mL round bottom flask, 0.20 g (0.69 mmol) of 3-isopropoxy-1- (2-methyl-4,5-dinitroimidazol-1-yl) propan-2-ol (**2**) and 1.38 mmol of primary amine in 7 mL of ethanol was mixed. It was allowed to stand for 30 min at room temperature and then 40 mL of cold water was dropped. The solution was extracted three times with methylene chloride. After evaporating off the solvent, the crude product obtained was crystallized from water.

*Appendix A.3. 3-Isopropoxy-1-(4-isobutylamino-2-metyl-5-nitroimidazol-1-yl)propan-2-ol (**3**)*

As a result, 0.125 g, (56.7% yield) of a yellow solid with mp = 120–122 °C was obtained; $R_f$ = 0.43 ($CH_2Cl_2$:$CH_3OH$ 9:1).

$^1$H NMR: δ 7.63 (s, 1H, NH), 5.43 (d, *J* = 2.8 Hz, 1H, OH), 4.02–4.11 (m, 1H, CH–OH), 3.83–3.90 (m, 2H, N–$CH_2$), 3.26–3.63 (m, 5H, $CH_2$–O, N–$CH_2CH(CH_3)_2$ and $OCH(CH_3)_2$), 2.27 (s, 3H, $CH_3$), 1.76–1.80 (m, 1H, N–$CH_2CH(CH_3)_2$), 1.11 (s, 6H, $OCH(CH_3)_2$), 0.91 (s, 6H, N–$CH_2CH(CH_3)_2$).

$^{13}$C NMR: δ 144.29 (C-4 Im), 141.64 (C-2 Im), 128.97 (C-5 Im), 71.42 (CH–OH), 69.62 (N–$CH_2$), 67.67 ($CH_2$–O), 50.78 ($OCH(CH_3)_2$), 48.23 (N–$CH_2CH(CH_3)_2$), 28.56 (N–$CH_2CH(CH_3)_2$), 21.91 ($OCH(CH_3)_2$), 19.49 (N–$CH_2CH(CH_3)_2$), 13.83 ($CH_3$).

MS *m/z* (%): 314.2 M$^+$ (54.6).

HRMS (ES): calcd. for $C_{14}H_{26}N_4O_4$: 314.38184, found: 314.38144.

IR: 3450 (ν N–H, w); 3270 (ν O–H, br); 2890 (ν C–H, m); 1524, 1326 (ν C–$NO_2$, m); 1485 (∂ C–H, m); 1385, 1365 (ν >$C(CH_3)_2$, m); 1250 (ν C–N, m); 1170 (ν C–O–C, m); 870 (ν C–H, m).

*Appendix A.4. 3-Isopropoxy-1-(4-phenylamino-2-methyl-5-nitroimidazol-1-yl)propan-2-ol (**4**)*

As a result, 0.02 g, (9.0% yield) of a yellow solid with mp = 72–75 °C was obtained. $R_f$ = 0.94 ($CH_2Cl_2$:$CH_3OH$ 9:1).

$^1$H NMR: δ 8.54 (s, 1H, NH), 7.17–7.21 (m, 2H, Ph), 6.84–6.87 (m, 1H, Ph), 6.69–7.71 (m, 2H, Ph), 5.18 (d, *J* = 5.3 Hz, 1H, OH), 3.22–3.95 (m, 6H, CH–OH, N–$CH_2$, $OCH(CH_3)_2$ and $CH_2$–O), 2.37 (s, 3H, $CH_3$), 1.05 (s, 6H, $OCH(CH_3)_2$).

$^{13}$C NMR: δ 142.35 (C-4 Im), 141.86 (C-2 Im), 132.95 (C-5 Im), 128.85 (Ph), 120.39 (Ph), 119.59 (Ph), 115.64 (Ph), 71.13 (CH–OH), 69.61 (N–$CH_2$), 68.21 ($CH_2$–O), 47.42 ($OCH(CH_3)_2$), 21.80 ($OCH(CH_3)_2$), 13.77 ($CH_3$).

MS *m/z* (%): 334.2 M$^+$ (34.1).

HRMS (ES): calcd. for $C_{16}H_{22}N_4O_4$: 334.37248, found: 334.37232.

IR: 3450 (ν N–H, w); 3270 (ν O–H, br); 3100 (ν Ar, w); 2890 (ν C–H, m); 1625–1575 (ν Ar, m); 1524, 1326 (ν C–NO$_2$, m); 1520 (ν C–N, s) 1485 (∂ C–H, m); 1385, 1365 (ν >C(CH$_3$)$_2$, m); 1250 (ν C–N, m); 1170 (ν C–O–C, m); 870 (ν C–H, m).

*Appendix A.5. 3-Isopropoxy-1-(2-methyl-4-(p-bromophenylamino)-5-nitroimidazol-1-yl)propan-2-ol (5)*

As a result, 0.05 g, (17.0% yield) of a yellow solid with mp = 56–59 °C was obtained; $R_f$ = 0.96 (CH$_2$Cl$_2$:CH$_3$OH 9:1).

$^1$H NMR: δ 8.66 (s, 1H, NH), 7.34 (s, 2H, Ph), 7.12 (s, 2H, Ph), 5.25 (d, *J* = 4,9 Hz, 1H, OH), 3.24–3.98 (m, 6H, CH–OH, N–CH$_2$, OCH(CH$_3$)$_2$ and CH$_2$–O), 2.37 (s, 3H, CH$_3$), 1.12 (s, 6H, OCH(CH$_3$)$_2$).

$^{13}$C NMR: δ 142.09 (C-4 Im), 135.60 (C-2 Im), 131.49 (Ph) 131.31 (C-5 Im), 117.43 (Ph), 115.73 (Ph), 111.34 (Ph), 71.21 (CH–OH), 69.66 (N–CH$_2$), 68.27 (CH$_2$–O), 47.43 (OCH(CH$_3$)$_2$), 21.85 (OCH(CH$_3$)$_2$), 13.81 (CH$_3$).

MS *m/z* (%): 413.8 M$^+$ (34,8).

HRMS (ES): calcd. for C$_{16}$H$_{21}$N$_4$O$_4$Br: 413.26864, found: 413.26831.

IR: 3450 (ν N–H, w); 3270 (ν O–H, br); 3100 (ν Ar, w); 2890 (ν C–H, m); 1625–1575 (ν Ar, m); 1524, 1326 (ν C–NO$_2$, m); 1520 (ν C–N, s) 1485 (∂ C–H, m); 1385, 1365 (ν >C(CH$_3$)$_2$, m); 1250 (ν C–N, m); 1170 (ν C–O–C, m); 870 (ν C–H, m); 610 (ν C–Br, m).

*Appendix A.6. 3-Isopropoxy-1-(2-methyl-5-nitro-4-morpholinoimidazol-1-yl)propan-2-ol (6)*

As a result, 0.10 g, (41.0% yield) of a yellow solid with mp = 126–129 °C was obtained; $R_f$ = 0.82 (CH$_2$Cl$_2$:CH$_3$OH 9:1).

$^1$H NMR: δ 5.27 (d, *J* = 4.9 Hz, 1H, OH), 3.78–4.10 (m, 3H, CH–OH, N–CH$_2$), 3.64–3.76 (m, 4H, 2xCH$_2$ 3,5-morpholine), 3.27–3.60 (m, 3H, CH$_2$–O, OCH(CH$_3$)$_2$), 2.99–3.19 (m, 4H, 2xCH$_2$ 2,6-morpholine), 2.32 (s, 3H, CH$_3$), 1.11 (s, 6H, OCH(CH$_3$)$_2$).

$^{13}$C NMR: δ 140.98 (C-4 Im), 140.69 (C-2 Im), 138.35 (C-5 Im), 69.81 (CH–OH), 68.32 (CH$_2$–O), 49.78 (N–CH$_2$), 49.57 (OCH(CH$_3$)$_2$), 47.94 (2xCH$_2$ 3,5-morpholine), 26.11 (2xCH$_2$ 3,5-morpholine), 21.94 (OCH(CH$_3$)$_2$), 13.93 (CH$_3$).

MS *m/z* (%): 328.3 M$^+$ (12.6).

HRMS (ES): calcd. for C$_{14}$H$_{24}$N$_4$O$_5$: 328.36516, found: 328.36546.

IR: 3330 (ν N–H, w); 3270 (ν O–H, br); 2910 (ν C–H, m); 1524, 1326 (ν C–NO$_2$, m); 1485 (∂ C–H, m); 1385, 1365 (ν >C(CH$_3$)$_2$, m); 1250 (ν C–N, m); 1170 (ν C–O–C, m); 870 (ν C–H, m).

*Appendix A.7. 3-Isopropoxy-1-(2-methyl-5-nitro-4-piperidinoimidazol-1-yl)propan-2-ol (7)*

As a result, 0.09 g, (40.0% yield) of a yellow solid with mp = 137–140 °C was obtained; $R_f$ = 0.83 (CH$_2$Cl$_2$:CH$_3$OH 9:1).

$^1$H NMR: δ 5.28 (d, *J* = 4.9 Hz, 1H, OH), 3.78–4.09 (m, 3H, CH–OH, N–CH$_2$), 3.27–3.60 (m, 3H, CH$_2$–O, OCH(CH$_3$)$_2$), 2.99–3.09 (m, 4H, 2xCH$_2$ 2,6-piperidine), 2.32 (s, 3H, CH$_3$), 1.56–1.62 (m, 6H, 3xCH$_2$ 3,4,5-piperidine), 1.11 (s, 6H, OCH(CH$_3$)$_2$).

$^{13}$C NMR: δ 140.99 (C-4 Im), 140.69 (C-2 Im), 138.35 (C-5 Im), 69.82 (CH–OH), 68.32 (CH$_2$–O), 49.78 (N–CH$_2$), 49.57 (OCH(CH$_3$)$_2$), 47.75 (2xCH$_2$ 2,6-piperidine), 25.65 (2xCH$_2$ 3,5-piperidine), 23.42 (CH$_2$ 4-piperidine), 21.94 (OCH(CH$_3$)$_2$), 13.93 (CH$_3$).

MS *m/z* (%): 326.2 M$^+$ (11.9).

HRMS (ES): calcd. for C$_{15}$H$_{26}$N$_4$O$_4$: 326.39284, found: 326.39271.

IR: 3330 (ν N–H, w); 3270 (ν O–H, br); 2910 (ν C–H, m); 1524, 1326 (ν C–NO$_2$, m); 1485 (∂ C–H, m); 1385, 1365 (ν >C(CH$_3$)$_2$, m); 1250 (ν C–N, m); 1170 (ν C–O–C, m); 870 (ν C–H, m).

*Appendix A.8. 3-Isopropoxy-1-(2-methyl-4-(N-methylpiperazino)-5-nitroimidazol-1-yl)propan-2-ol (8)*

As a result, 0.10 g, (40.0% yield) of a yellow solid with mp = 74–76 °C was obtained; $R_f$ = 0.2 (CH$_2$Cl$_2$:CH$_3$OH 9:1).

$^1$H NMR: δ 5.29 (d, *J* = 4.9 Hz, 1H, OH), 3.76–4.08 (m, 3H, CH–OH, N–CH$_2$), 3.47 (t, *J* = 5.6 Hz, 4H, 2xCH$_2$ 2,6-piperazine), 3.27–3.62 (m, 3H, CH$_2$-O, OCH(CH$_3$)$_2$), 2.42

(t, *J* = 4.8 Hz, 4H, 2xCH$_2$ 3,5-piperazine), 2.35 (s, 3H, N–CH$_3$), 2.32 (s, 3H, CH$_3$), 1.12 (s, 6H, OCH(CH$_3$)$_2$).

MS *m/z* (%): 341.2 M$^+$ (7.9).

HRMS (ES): calcd. for C$_{15}$H$_{27}$N$_5$O$_4$: 341.40768, found: 341.40731.

IR: 3330 ($\nu$ N–H, w); 3270 ($\nu$ O–H, br); 2910 ($\nu$ C–H, m); 1524, 1326 ($\nu$ C–NO$_2$, m); 1485 ($\partial$ C–H, m); 1385, 1365 ($\nu$ >C(CH$_3$)$_2$, m); 1250 ($\nu$ C–N, m); 1170 ($\nu$ C–O–C, m); 870 ($\nu$ C–H, m).

### Appendix A.9. General Method for Reaction with α-Aminoacids

In a 100 mL round bottom flask, 0.2 g (0.69 mmol) of 3-isopropoxy-1-(2-methyl-4,5-dinitroimidazol-1-yl)propan-2-ol (**2**), 2.07 mmol of α-aminoacid and 10 mL of 75% ethanol was mixed. The mixture was heated to reflux in a water bath for 30 min, then the solvent was distilled off using a rotary vacuum evaporator. The residue was purified by gravity column chromatography on a silica gel column with CHCl$_3$:MeOH (9:1) mixture as the eluent. The crude product was obtained by slowly evaporating the solvent from the collected fractions. It was then crystallized from methanol to give a yellow, crystalline solid.

### Appendix A.10. 2-{N-[1-(3-Isopropoxy-2-hydroxypropyl)-2-methyl-5-nitroimidazol-4-yl]}amino-4-methylthiobutanoic Acid (**9**)

As a result, 0.03 g, (11.0% yield) of a yellow solid with mp = 55–57 °C was obtained; $R_f$ = 0.87 (CH$_2$Cl$_2$:CH$_3$OH 9:1).

$^1$H NMR: δ 8.45 (s, 1H, NH), 5.33–5.55 (m, 1H, OH), 3.25–4.22 (m, 7H, N–CH$_2$, CH$_2$–O, CH–OH, OCH(CH$_3$)$_2$, CH–COOH), 1.93–2.70 (m, 10H, S–CH$_3$, CH$_2$CH$_2$–S Met, CH$_3$), 1.12 (s, 6H, OCH(CH$_3$)$_2$).

$^{13}$C NMR: δ 167.21 (C = O), 147.06 (C-4 Im), 139.85 (C-2 Im), 128.44 (C-5 Im), 59.63 (CH–COOH Met), 67.38 (CH–OH), 60.78 (N–CH$_2$), 48.31 (CH$_2$–O), 48.15 (OCH(CH$_3$)$_2$), 31.11 (CH$_2$CH$_2$–S Met), 31.05 (CH$_2$CH$_2$–S Met), 21.93 (OCH(CH$_3$)$_2$), 14.62 (S–CH$_3$ Met), 13.88 (CH$_3$).

MS *m/z* (%): 390.3 M$^+$ (3.9).

HRMS (ES): calcd. for C$_{15}$H$_{26}$N$_4$O$_6$S: 390.45684, found: 390.45676.

IR: 3330 ($\nu$ N–H, w); 2920 ($\nu$ O–H, br); 2910 ($\nu$ C–H, m); 1620 ($\nu$ C = O, s) 1524, 1326 ($\nu$ C–NO$_2$, m); 1485 ($\partial$ C–H, m); 1385, 1365 ($\nu$ >C(CH$_3$)$_2$, m); 1250 ($\nu$ C–N, m); 1170 ($\nu$ C–O–C, m); 870 ($\nu$ C–H, m).

### Appendix A.11. 2-{N-[1-(3-Isopropoxy-2-hydroxypropyl)-2-methyl-5-nitroimidazol-4-yl]}amino-3-methylobutanoic Acid (**10**)

As a result, 0.02 g, (9.0% yield) of a yellow solid with mp = 150–152 °C was obtained; $R_f$ = 0.77 (CH$_2$Cl$_2$:CH$_3$OH 9:1).

$^1$H NMR: δ 8.43 (s, 1H, NH), 5.33–5.55 (m, 1H, OH), 3.25–4.22 (m, 7H, N–CH$_2$, CH$_2$-O, CH–OH, OCH(CH$_3$)$_2$, CH–COOH), 2.49–2.54 (m, 1H, CH(CH$_3$)$_2$ Val), 2.25 (s, 3H, CH$_3$), 1.01–1.13 (m, 12H, OCH(CH$_3$)$_2$, CH(CH$_3$)$_2$ Val).

$^{13}$C NMR: δ 153.23 (C = O), 145.01 (C-4 Im), 141.68 (C-2 Im), 129.19 (C-5 Im), 69.63 (CH–COOH Val), 67.38 (CH–OH), 61.73 (N–CH$_2$), 48.32 (CH$_2$–O), 48.13 (OCH(CH$_3$)$_2$), 31.06 (CH(CH$_3$)$_2$ Val), 21.93 (OCH(CH$_3$)$_2$), 21.90, 18.58 (CH(CH$_3$)$_2$ Val), 13.89 (CH$_3$).

MS *m/z* (%): 358.3 M$^+$ (3.4).

HRMS (ES): calcd. for C$_{15}$H$_{26}$N$_4$O$_6$: 358.39084, found: 358.39079.

IR: 3330 ($\nu$ N–H, w); 2920 ($\nu$ O–H, br); 2910 ($\nu$ C–H, m); 1620 ($\nu$ C = O, s) 1524, 1326 ($\nu$ C–NO$_2$, m); 1485 ($\partial$ C–H, m); 1385, 1365 ($\nu$ >C(CH$_3$)$_2$, m); 1250 ($\nu$ C–N, m); 1170 ($\nu$ C–O–C, m); 870 ($\nu$ C–H, m).

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
