# Peer review of "Synthesis and Predicted Activity of Some 4-Amine and 4-(α-Aminoacid) Derivatives of N-Expanded-metronidazole Analogues"

_compounds, doi:10.3390/compounds3010009_

Round 1

Reviewer 1 Report

In this manuscript, the authors reported the synthesis of a series of new 4-amine-5-nitroimidazole derivatives with similar structures to metronidazole, a type of antibacterial drug, and analyzed biological activity of eight compounds. This is an interesting study and subsequently deserves publication in Compounds, however after addressing the following concerns.

1. For the characterization in appendix, I was wondering if the authors could obtain the purity of final products through NMR or other methods.  

2.  In the reference part, the issue numbers are included in some of publications. Please keep the format consistent according to the journal requirements.

Author Response

Dear Editor,

Dear Reviewers,

               Thank you very much for the detailed and polite review of the article. I agree with all the allegations and inaccuracies. Below is a list of the corrections made, according to the suggestions of Reviewers:

Reviewer 1:

  •   The purity of final products were obtain through TLC, melting points and spectral methods;
  •   The format of citing references has been checked and needed corrections has been made.

 Justyna Żwawiak

 /corresponding author/

Reviewer 2 Report

In the present study, the authors have synthesized a number of new derivatives of 3-Isopropoxy-1-(2-methyl-4,5-di-nitroimidazol-1-yl)propan-2-ol where nitro-group in 4th position is substituted with some primary and secondary amines as well as alpha-aminoacids (methionine and valine). These nitroimidazole derivatives have been characterized using IR-, 1H and 13C NMR spectroscopy, mass- and high resolution mass spectrometry (MS and HRMS). Also, all nitroimidazole derivatives were analysed by the PASS program and AntiBac-Pred methods.

The present manuscript is a development of works performed by the authors during the last ~30 years. In general, I believe the results may attract the attention of the reading audience of the journal Compounds. Before the acceptance some revisions are necessary:

  1. All compounds (2-10) were characterized by 1H and 13C NMR spectroscopy. The corresponding spectroscopic details are given in Appendix A. However, it is necessary to include the figures of the NMR spectra for all new compounds in the Electronic Supplementary Information file (ESI).
  2. Some fragmented IR data are present in the characterization of compounds. However, in the present state, it is not meaningful.  For example, compounds 3, 6, 7, and 8 have the identical description: “IR: 3400-3500 (ν O–H); 1524, 1326 (ν C–NO2); 1020-1250 (ν C–N).” The selected bands for other compounds are also too similar. One more question: why a range 1260-1340 cm-1 was chosen as a range for ν C–N vibrations in IR of compounds 24, and 5, but the same bond vibrations (ν C–N) for compounds36-10 are given in the range 1020-1250 cm-1?  The bonds C–N in these compounds have the same nature and the close environment, but their ranges of vibrations are so different that they don't even overlap (1020-1250 and 1260-1340 cm-1)!!! Please, check for consistency.
  3. The IR spectral data should be given as a full set of vibrations with an indication of strong (s), medium (m), or weak (w) bonds. E.g. “… 1750 s, 1250 s, 1215 w, 1130 m, …”
  4. The figures with IR spectra also have to be shown in ESI for all compounds.
  5. Small moment: for compounds 9 and 10, the vibration “1750 (ν C–O)” should be “1750 (ν C=O)”.
  6. Some concise conclusions and perspectives are expected for this manuscript.

In summary, the ESI with NMR and IR spectral data are absolutely necessary for this manuscript. It should be provided. Also, the authors need to add concise conclusions.

Author Response

Dear Editor,

Dear Reviewers,

               Thank you very much for the detailed and polite review of the article. I agree with all the allegations and inaccuracies. Below is a list of the corrections made, according to the suggestions of Reviewers:

Reviewer 2:

  • IR data has been improved;
  • Small editorial mistakes has been corrected;
  • Conclusion has been added;
  • NMR and IR spectra were of course recorded and they confirm the obtained structures, but I must admit that including them is a problem for us. The reason is that they were performed over many years, starting in the early 2000s, and we do not have electronic versions of them. The "paper" versions that we have were treated "working" and they bear traces of many thoughts marked by hand, and of course concerning the assignment of individual signals to the appropriate atoms of the molecule... Nevertheless, the final quality of this type of paper versions of these spectra is simply not good to be publishing in a MDPI journal. I hope for your understanding.

Best wishes,

Justyna Żwawiak

 /corresponding author/

Reviewer 3 Report

In the current manuscript, the authors synthesized 4-amine and 4-(α-Aminoacid)-derivatives and predicted their activity using PASS and AntiBac-Pred methods.

The abstract section has summarized major aspects of the paper such as the key purpose of the study, research problems, basic design of the study, and key findings. The introduction part is well written by illustrating the background and significance of 4-amine-5-nitroimidazole derivatives.

The manuscript was clearly written and all the necessary information such as reaction scheme and experimental procedures were provided. Product characterization was mainly done using 1H and 13C NMR.  

Biological activity of synthesized compounds was predicted using Pa values

The manuscript is missing conclusion. Author needs to provide conclusion to summarize the work conducted, results obtained, and conclusions drawn based on the results.

I would recommend the manuscript for a minor revision.

Author Response

Dear Editor,

Dear Reviewers,

               Thank you very much for the detailed and polite review of the article. I agree with all the allegations and inaccuracies. Below is a list of the corrections made, according to the suggestions of Reviewers:

Reviewer 3:

Conclusion has been included in the manuscript.

Justyna Żwawiak

 /corresponding author/

Round 2

Reviewer 2 Report

Dear authors,

thank you very much for your corrections. But I insist to see the IR spectra and other evidences that the compounds, you want to be reported, were really obtained. The spectra should not be very beautiful and they should not be for publication. It is no matter what form they have (photos, scans and so on), but they must be present just for evaluation. The list of stretching vibrations it not enough. 

And the second point. Are the data still up to date, if the synthesis was performed about 20 years ago?   

I am ready to evaluate the revised manuscript with all the actual data, but now to my regret I can not recommend it for publication. 

Author Response

Dear Reviewer,

I am sending the spectra of compounds.

Regards,

Authors

Round 3

Reviewer 2 Report

Dear authors,

Thank you for your efforts.